# The Lipid Paradox Among Acute Ischemic Stroke Patients-A Retrospective Study of Outcomes and Complications

**DOI:** 10.3390/medicina55080475

**Published:** 2019-08-13

**Authors:** Urvish Patel, Preeti Malik, Mihir Dave, Matthew S. DeMasi, Abhishek Lunagariya, Vishal B. Jani, Mandip S. Dhamoon

**Affiliations:** 1Department of Neurology, Creighton University School of Medicine, Omaha, NE 68124, USA; 2Department of Public Health, Icahn School of Medicine at Mount Sinai, New York, NY 10029, USA; 3Department of Internal Medicine, University of Nevada Reno School of Medicine, Reno, NV 89102, USA; 4Department of Internal Medicine, Albert Einstein College of Medicine, Bronx, NY 10461, USA; 5Department of Neurology, Icahn School of Medicine at Mount Sinai, New York, NY 10029, USA

**Keywords:** stroke, hyperlipidemia, epilepsy, stroke associated pneumonia, gastro-intestinal hemorrhage, hemorrhagic transformation, nationwide inpatient sample, outcomes

## Abstract

Background and objectives: The Studies have suggested hypercholesterolemia is a risk factor for cerebrovascular disease. However, few of the studies with a small number of patients had tested the effect of hypercholesterolemia on the outcomes and complications among acute ischemic stroke (AIS) patients. We hypothesized that lipid disorders (LDs), though risk factors for AIS, were associated with better outcomes and fewer post-stroke complications. Materials and Method: We performed a retrospective analysis of the Nationwide Inpatient Sample (years 2003–2014) in adult hospitalizations for AIS to determine the outcomes and complications associated with LDs, using ICD-9-CM codes. In 2014, we also aimed to estimate adjusted odds of AIS in patients with LDs compared to patients without LDs. The multivariable survey logistic regression models, weighted to account for sampling strategy, were fitted to evaluate relationship of LDs with AIS among 2014 hospitalizations, and outcomes and complications amongst AIS patients from 2003–2014. Results and Conclusions: In 2014, there were 28,212,820 (2.02% AIS and 5.50% LDs) hospitalizations. LDs patients had higher prevalence and odds of having AIS compared with non-LDs. Between 2003–2014, of the total 4,224,924 AIS hospitalizations, 451,645 (10.69%) had LDs. Patients with LDs had lower percentages and odds of mortality, risk of death, major/extreme disability, discharge to nursing facility, and complications including epilepsy, stroke-associated pneumonia, GI-bleeding and hemorrhagic-transformation compared to non-LDs. Although LDs are risk factors for AIS, concurrent LDs in AIS is not only associated with lower mortality and disability but also lower post-stroke complications and higher chance of discharge to home.

## 1. Introduction

Hypercholesterolemia is a well-documented risk factor for cardiovascular morbidity and mortality [1,2,3,4]. However, the relationship between ischemic stroke and cholesterol is complex and appears to contain several paradoxes [5,6,7,8,9,10,11,12]. Many large-scale studies on stroke and cholesterol have not differentiated between ischemic and hemorrhagic stroke, nor among various subtypes of ischemic stroke [6,8,9,10]. It has been demonstrated that cholesterol may increase the risk of only certain types of stroke [13,14,15], and low cholesterol levels predispose to hemorrhagic stroke [10,16], thus weakening an association between cholesterol and all stroke. Regardless of increased cardiovascular disease risk with high cholesterol levels and decreased stroke risk with statin use, a higher cholesterol value has been associated with a better stroke outcome in several studies [17,18,19,20]. Further research has shown “reverse epidemiology” between cholesterol levels at admission, statin treatment and stroke morbidity. Earlier studies have shown a positive association between elevated admission cholesterol at ischemic stroke onset and improved short-term functional outcome [21] and 10-year survival [22].

These paradoxical observations may be related to differential associations with different stroke subtypes. Several recent studies reported that cholesterol was lowest in cardioembolic strokes [15,23,24]. The largest prospective cohort of 4128 adults aged >70 years reported that high or normal/borderline total cholesterol values were associated with good survival compared to low values [25].

We aimed to estimate odds of having AIS with LDs, and whether or not LDs in AIS patients were associated with better outcomes.

## 2. Materials and Methods

Data was obtained from the Nationwide Inpatient Sample (NIS) between January 2003 and December 2014. The NIS is the largest publicly available all-payer inpatient care database in the United States and contains discharge-level data provided by states that participate in the HCUP (including a total of 46 in 2011). This administrative dataset contains data on approximately 8 million hospitalizations in 1000 hospitals that were chosen to approximate a 20% stratified sample of all US community hospitals, representing more than 95% of the national population. Criteria used for stratified sampling of hospitals into the NIS include hospital ownership, patient volume, teaching status, urban or rural location, and geographic region. Discharge weights are provided for each patient discharge record, which allow extrapolation to obtain national estimates. Each hospitalization is treated as an individual entry in the database and is coded with one principal diagnosis, up to 24 secondary diagnoses, and 15 procedural diagnoses associated with that stay. Detailed information on NIS is available at http://www.hcup-us.ahrq.gov/db/nation/nis/nisdde.jsp.

The data were taken from the Nationwide Inpatient Sample, which is a deidentified database from “Health Care Utilization Project (HCUP)” sponsored by the Agency for Healthcare Research and Quality, USA, thus informed consent or IRB approval was not needed for the study. The relevant ethical oversight and HCUP Data Use Agreement (HCUP-4Q28K90CU) were obtained for the study.

### 2.1. Study Population 

We used the ninth revision of the International Classification of Diseases, clinical modification codes (ICD-9-CM) to identify adult patients admitted with a primary diagnosis of AIS (ICD-9-CM codes 433.01, 433.11, 433.21, 433.31, 433.81, 433.91, 434.01, 434.11, 434.91). These codes have been previously validated and are 35% sensitive, 99% specific, 96% positive predictive value (PPVs), and 79% negative predictive value for the diagnosis of ischemic stroke [26]. Similarly, patients with secondary diagnosis of LDs were identified using ICD-9-CM codes 272.0, 272.1, and 272.2 (sensitivity of 27.0%, specificity of 76.7%, PPV of 71.1%, and NPV of 33.1%) [27]. We used ICD-9-CM codes to identify independent predictors (covariates), including the comorbidities of hypertension, diabetes mellitus, hypercholesterolemia, atrial fibrillation, use of anticoagulant and antiplatelet medications, chronic use of NSAIDs and aspirin, smoking (current/past), use of IV tPA, drug abuse, alcohol dependence, smoking status, mechanical thrombectomy, AV malformation, amyloidosis, atrial fibrillation, nasogastric tube, gastrostomy, endotracheal intubation, non-invasive mechanical ventilation, invasive mechanical ventilation and H. pylori infection. Appendix A lists all ICD-9-CM codes that were used for this study. 

AIS patients were stratified by LDs status. Age < 18 years and admissions with missing data for age, sex, and race were excluded. 

### 2.2. Patient and Hospital Characteristics

Patient characteristics of interest were sex, age, race, insurance status and concomitant diagnoses as defined above. Race was defined by White (referent), African American, Hispanic, Asian or Pacific Islander, and Native American. Insurance status was defined by Medicare (referent), Medicaid, private insurance, and other/self-pay/no charge. We defined the severity of co-morbid conditions using Deyo’s modification of the Charlson co-morbidity index (CCI) (Appendix A). Thirty-one facilities were considered to be teaching hospitals if they have an American Medical Association–approved residency program, are a member of the Council of Teaching Hospitals, or have a full-time equivalent interns and residents to patient’s ratio of ≥0.25. HCUP NIS contains data on total charges for each hospital in the databases, which represents the amount that hospitals billed for services. 

### 2.3. Outcomes

We tested for associations between LDs and AIS amongst the January 2014–December 2014 dataset. We also examined outcomes such as all-cause in hospital mortality, All Patient Refined Diagnosis Related Groups (APR-DRG) risk of death (RoD), APR-DRG loss of function (LoF), discharge disposition (DD) [home vs. transfer to short-term hospital (STH), skilled nursing facility (SNF), intermediate care facility (ICF)], post-stroke complications like epilepsy, stroke-associated pneumonia (SAP), hemorrhagic transformation (HT), upper gastro-intestinal bleeding (UGIB), length of stay (LoS), and cost of hospitalization with LDs amongst AIS hospitalizations (years 2003–2014). The comparison of disability/loss of function was investigated by APR-DRGs severity between patients with LDs and patients without LDs on discharge. Similarly, risk of death was assigned using APR-DRG likelihood of death (risk of death) on discharge. APR-DRGs were assigned using software developed by 3M Health Information Systems, where score 1 indicates minor, 2—moderate, 3—major, 4—extreme loss of function or likelihood of death on discharge.

### 2.4. Statistical Analysis

All statistical analyses were performed using the weighted survey methods in SAS (version 9.4). Weighted values of patient-level observations were generated to produce a nationally representative estimate of the entire US population of hospitalized patients. Univariate analysis of differences between categorical variables was tested using the chi-square test and analysis of differences between continuous variables (LoS and cost of hospitalization) was tested using Student’s t-test. Mixed-effects survey logistic regression models with weighted analysis were used for the categorical dependent variables, including LDs and outcomes of interest, in order to estimate odds ratio (OR) and 95% confidence intervals for the association between AIS and LDs in the 2014 cohort as well as LDs and outcomes amongst AIS hospitalizations during years 2003–2014. 

The hierarchical models (demographics and patient-level factors nested within hospital-level factors) were created as random effects within the model for the outcomes. In the multivariate models, we had included demographics (age, gender, race), patient-level hospitalization variables (admission day, primary payer, admission type, median household income category), hospital-level variables (hospital region, teaching versus nonteaching hospital, hospital bed size), comorbidities/concurrent conditions like hypertension, diabetes mellitus, hypercholesterolemia, atrial fibrillation, obesity, hemorrhagic transformation, smoking status, drug abuse, alcohol abuse, medication use (anticoagulant and antiplatelet medication, chronic use of aspirin), and use of IV tPA during the same hospitalization or in a different institution within the 24 h prior to admission to the facility, mechanical thrombectomy, gastrostomy, nasogastric tube insertion, invasive-noninvasive mechanical ventilation, and CCI. The confounders of the models were tailored according to the need of the individual model.

We investigated the link between LDs and AIS, LDs and post-AIS outcomes, and LDs and post-AIS complications by creating separate mix effect survey logistic regression models with weights to account for sampling strategy to find out: 

(1) Relationship of LDs with AIS amongst year 2014 hospitalizations;

(2) Relationship of LDs with post-AIS outcomes and complications amongst AIS hospitalizations from year 2003–2014;

Model 1: All cause in hospital mortality;

Model 2: Discharge disposition (home vs. non-home);

Model 3: APR-DRG loss of function (major/severe vs. minor/moderate);

Model 4: APR-DRG risk of death (major/severe likelihood vs. minor/moderate likelihood);

Model 5: Post-AIS early epilepsy;

Model 6: Stroke Associated Pneumonia (SAP);

Model 7: Upper gastro-intestinal bleeding (UGIB);

Model 8: Hemorrhagic transformation (HT).

For each model, C-index was calculated. All statistical tests used were 2-sided, and *p* < 0.05 was deemed statistically significant.

## 3. Results

### 3.1. LDs and AIS Amongst Year-2014 Hospitalizations

We identified 28,212,820 total hospitalizations in 2014, (Figure 1A) out of which (569,215) 2.02% and (1,550,956) 5.50% patients were hospitalized with primary or secondary diagnosis of AIS and LDs respectively. Out of 569,215 patients with AIS, 50,005 (8.78%) had LDs. Patients with LDs had higher prevalence of having AIS (3.22% vs. 1.95%; *p* < 0.0001) compared with non-LDs (Table 1).

Table 2 includes regression models among 2014 hospitalizations with odds of having AIS amongst LDs vs. non-LDs after adjusting for patients’ demographics, patients and hospital level characteristics, co-morbidities, and CCI. In multivariate survey logistic regression analysis, LD was associated with higher adjusted odds of having AIS (aOR: 1.18; 95% CI:1.15–1.20; *p* < 0.0001) compared to non-LDs. The AUC or C statistic of the ROC was used to validate the accuracy of the regressions. The AUC was 0.88, which indicates a precise/good model.

### 3.2. LDs and Post-AIS Outcomes Amongst AIS Population from Year 2003–2014

We found a total of 4,224,924 hospitalizations due to AIS from year 2003 to 2014 after excluding patients with age < 18 years and admissions with missing data for age, gender, and race (Figure 1B). Out of 4,224,924 AIS hospitalizations, 451,645 (10.69%) had LDs. 

We analyzed prevalence trends of LDs in AIS hospitalizations. As shown in Figure 2, trends of LDs in AIS hospitalizations were slightly declining from years 2003 to 2014. LDs prevalence percentage declined from 12.17% in 2003 to 9.31% in 2014 (*p*-_Trend_ < 0.0001).

AIS hospitalizations with LDs were more likely to be female (49.35% vs. 46.85%, *p* < 0.0001). There was no difference in other demographic characteristics like age and race among AIS hospitalizations with and without hyperlipidemia. Co-morbidities such as diabetes (40.02% vs. 33.50%, *p* < 0.0001), hypertension (86.44% vs. 78.66%, *p* < 0.0001), and obesity (9.85% vs. 7.63%, *p* < 0.0001) were higher in patients with LDs than those without LDs. AIS hospitalizations in large, urban (teaching and non-teaching) hospitals and in the Northeast region were more likely to have patients with LDs (Table 3). 

Table 4 includes outcomes of LDs among AIS hospitalizations. Our outcomes of interest were to identify post-AIS outcomes (in-hospital mortality, discharge, disability, risk of death) and complications (post stroke early epilepsy, stroke associated pneumonia, hemorrhagic transformation and upper gastro-intestinal bleeding) amongst AIS hospitalizations. All cause in-hospital mortality was lower in AIS with LDs (2.93% vs. 5.48%, *p* < 0.0001) than without LDs. Some 43.14% of patients with LDs had discharged to home compared to 36.81% without LDs (*p* < 0.0001). Overall, AIS hospitalizations with LDs had a higher prevalence of discharge to home or routine (43.14% vs. 36.81%, *p* < 0.0001) and a lower prevalence of discharge other than home (56.86% vs. 63.19%, *p* < 0.0001) compared to patients without LDs. The prevalence of major/severe loss of function was lower (29.92% vs. 37.75%, *p* < 0.0001) among AIS hospitalizations with LDs than without LDs. The patients with LDs was also associated with lower prevalence major/extreme likelihood of death (15.8% vs. 21.89%, *p* < 0.0001) in AIS hospitalizations. Prevalence of post stroke early epilepsy (4.68% vs. 6.13%, *p* < 0.0001), SAP (2.14% vs. 3.70%, *p* < 0.0001), HT (1.25% vs. 1.71%, *p* < 0.0001) and UGIB (0.33% vs. 0.45%, *p* < 0.0001) were lower among AIS hospitalizations with LDs than without LDs.

Mean length of stay (4.83 days vs. 5.83 days, *p* < 0.0001) and total cost of hospitalization were lower amongst patients with LDs ($34,604 vs. 38,547, *p* < 0.0001) (Table 4).

Table 5 lists multivariate analysis of outcomes and complications in AIS hospitalizations. LDs were associated with lower adjusted odds of all cause in-hospital mortality (aOR: 0.66, 95% CI: 0.62–0.69, *p* < 0.0001), discharge disposition (home vs. no-home) (aOR: 0.83, 95% CI: 0.82–0.85, *p* < 0.0001), APR-DRG loss of function (major/severe vs. minor/moderate) (aOR: 0.80, 95% CI: 0.79–0.82, *p* < 0.0001), and APR-DRG risk of death (major/severe likelihood vs. minor/moderate likelihood) (aOR: 0.77, 95% CI: 0.75–0.79, *p* < 0.0001) in comparison to patients without LDs amongst AIS hospitalizations of year 2003–2014 (Models 1–4).

LDs were associated with lower adjusted odds of post stroke early epilepsy (aOR:0.89, 95% CI: 0.8-0.86, *p* < 0.0001), SAP (aOR: 0.75, 95% CI: 0.71–0.80, *p* < 0.0001), upper GI bleeding (aOR:0.85, 95% CI: 0.73–0.99, *p* < 0.0001) and hemorrhagic transformation (aOR: 0.82, 95% CI: 0.75–0.89, *p* < 0.0001) in comparison to patients without LDs (Models 5–8).

C statistic was used to validate the accuracy of the regressions. All models have c-index >0.6, which indicates a good model fit (Table 5).

## 4. Discussion

In this study, we aimed to investigate the link between acute ischemic stroke (AIS) and lipid disorders (LDs), specifically the odds of having AIS with LDs, and whether LDs were associated with better outcomes or less complications in AIS patients as compared to those without LDs. We did this by performing a population-based retrospective cross-sectional analysis of the NIS in adult hospitalizations for AIS.

Our study found that those with LDs had a higher prevalence of being hospitalized with AIS as compared to those without LDs. Co-morbidities, such as diabetes, hypertension, obesity, and depression were also higher in those with LDs than in those without them. Our findings are similar to a study by Olsen et al. who have reported that patients with hypercholesterolemia are at increased risk of stroke [22]. However, the paradoxical relationship is evident when analyzing for post-AIS outcomes such as in-hospital mortality, discharge status, disability, and risk of death, as well as post-AIS complications such as post-stroke early epilepsy, stroke associated pneumonia, hemorrhagic transformation, and upper GI bleeding. 

For AIS patients with lipid disorders, there was reduced in-hospital mortality (aOR: 0.66, *p* < 0.0001), increased discharge to home (aOR: 0.83, *p* < 0.0001), decreased major/severe loss of function (aOR: 0.80, *p* < 0.0001), and decreased major/extreme likelihood of death (aOR: 0.77, *p* < 0.0001); additionally, there was reduced risk of post stroke epilepsy (aOR:0.89, *p* < 0.0001), SAP (aOR: 0.75, *p* < 0.0001), upper GI bleeding (aOR:0.85, *p* < 0.0001) and hemorrhagic transformation (aOR: 0.82, *p* < 0.0001) as compared to AIS patients without LDs. There are supporting studies to our paradoxical findings that have reported higher cholesterol levels on admission are associated with better long-term survival or outcomes amongst AIS survivors [22,28,29].

There is thus an incongruity that is seen in AIS patients; on one hand, there is a benefit to reducing cholesterol and LDs in that there is a reduced prevalence of AIS and co-morbidities as compared to those without LDs, as well as for being beneficial for cerebrovascular mortality and morbidity [3,4]. However, patients with LDs have better post-AIS outcomes and less post-AIS complications. 

This paradox may be explained by the type of stroke that is often seen with those with LDs. Those with LDs may be more predisposed toward small-vessel strokes, and thus will have less severe strokes with better prognoses; as a result, they would have improved post-AIS outcomes and reduced post-AIS complications as compared to those without LDs who may be more prone to the often more severe large brain vessel occlusion. This theory can be supported by our data, as well as that large occlusion of brain vessels as seen in cardio embolic strokes have the lowest total serum cholesterol levels [23,24].

Interestingly, the trends of LDs in AIS hospitalizations have decreased from 2003–2014. This may be due to reduced hospitalizations for minor strokes and may support theory that those with LDs are more prone to small-vessel strokes which have better prognoses. Thus, those patients with LDs and minor strokes are often not being hospitalized as much, and if they are, they will show improved post-AIS outcomes and reduced complications as shown in our study. However, in our study AIS was analyzed as a whole, and was not classified into subtypes using TOAST classification. This was one limitation of our study to support the decreasing trends and theory of LDs association with small vessel stroke which have better prognoses. 

This analysis and confirmation of the paradoxical relationship seen between LDs and AIS may have implications for treatment strategies in the future. Statins are certainly beneficial to cardiovascular health, have been proven to have neuroprotective and microvascular benefits in animal stroke models, and may be important in augmenting cerebral repair after ischemic injury [20]. They also have been proven to improve survival for up to a year after stroke.

Further work must be done in order to fully determine why LDs seem to have a beneficial effect on AIS outcomes and complications, and to determine potential treatment strategies. Clinical trials should be run to determine the effects of LDs and statins on AIS, AIS subtypes such as small-vessel disease, and AIS outcomes and complications.

A major strength of the study was that findings were nationally representative for the USA. NIS data is a large deidentified inpatient database, and our study has good statistical power. The APR-DRG coding systems used in this study to assess the severity of illness and risk of mortality are externally validated. They are very reliable and consistent, and widely used by hospitals, consumers, payers and regulators [30,31].

One main limitation to this study mentioned above, AIS was not classified into subtypes using TOAST classification (such as cardio-embolic, small-vessel, or large artery atherothrombotic strokes). Further investigations should classify AIS into these subtypes and determine if there is an association between LDs and these subtypes. This would be able to determine if theory that LDs cause the less severe small-vessel strokes is correct, and thus is the reason why we see better AIS-outcomes and reduced complications with LDs. Another limitation of the study was LDs were identified as a secondary diagnosis (history of LDs or first-time diagnosis) responsible for AIS and we did not have data on statin use prior or during the hospitalization. So neuroprotective role of statin [20] was difficult to differentiate. Other limitations to this study include the fact that we only considered patients who were hospitalized with strokes; between 10%–40% of stroke patients are not admitted to the in-patient hospital [13]. Thus, patients with less severe strokes may not be included in this study. The outcomes evaluated while patients were in hospital and status of patients on discharge, we had not evaluated the long-term outcomes. Additionally, although co-morbidities were accounted for and adjusted for in analysis, other subtler differences between patients such as prior strokes or other unaccounted medications may flaw our design outcomes. Though we had adjusted outcomes model with antiplatelets or anticoagulant use before and during hospitalization, but we had no record on statin use to adjust the models.

## 5. Conclusions

LDs have been shown to have an increased prevalence in patients with AIS hospitalizations as compared to those without LDs. However, we have also confirmed that LDs have paradoxically been shown to improve post-AIS outcomes and reduce post-AIS complications. This complicated relationship between stroke and LDs requires further work and research to determine the reasoning as to the associated benefit between higher lipids and better outcomes, and whether the theory in that LDs have a stronger association in causing small vessel and thus less severe strokes with better prognoses can be confirmed. Clinical trials should be undertaken to further determine the relationship of statin uses and outcomes to refine treatment strategies before and after stroke onset, and further work should be initiated to determine the association between LDs and AIS subtypes such as small-vessel strokes, AIS-subtype outcomes, and AIS-subtype complications.

## Figures and Tables

**Figure 1 medicina-55-00475-f001:**
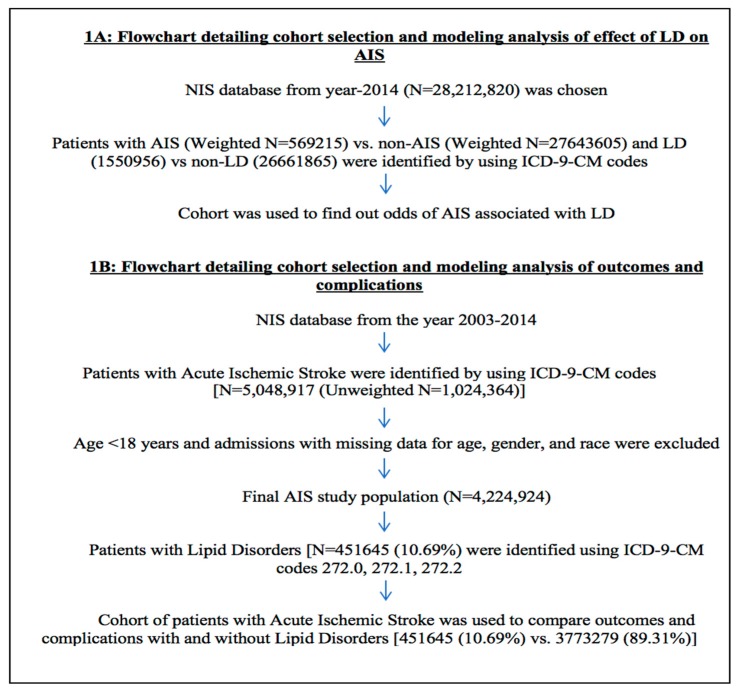
Flowchart detailing cohort selection and analysis modeling.

**Figure 2 medicina-55-00475-f002:**
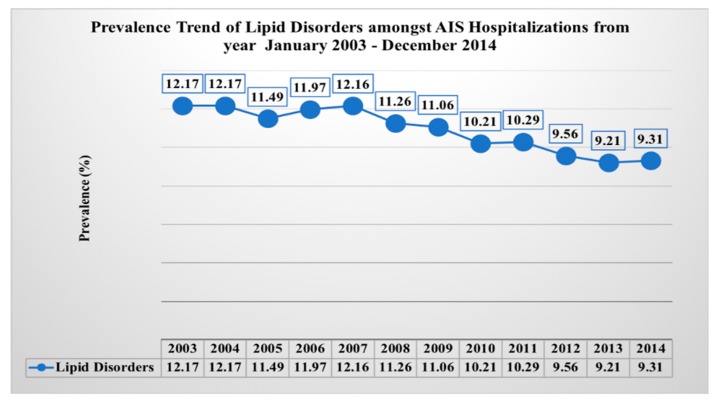
Prevalence trend of lipid disorders.

**Table 1 medicina-55-00475-t001:** Univariate associations of acute ischemic stroke (AIS) with lipid disorders (LDs) in the year 2014.

	LD	No LD	Total	*p* Value
AIS	50,005 (3.22%)	519,210 (1.95%)	569,215	<0.0001
No AIS	1,500,951 (96.78%)	26,142,654 (98.05%)	27,643,605	
	1,550,956	26,661,864	28,212,820	

**Table 2 medicina-55-00475-t002:** Multivariate logistic regression analysis of predictors of AIS and associations with LDs among year 2014 hospitalizations.

	OR	95% Confidence Limits		*p* Value
		LL	UL	
**No Lipid Disorders**	Reference			
**Lipid Disorders**	1.18	1.15	1.20	<0.0001
**Demographics of Patients**				
**Age (Years)**	1.02	1.02	1.02	<0.0001
**Gender**				
Female	Reference			
Male	1.00	0.99	1.01	0.8272
**Race**				
White	Reference			
African American	1.17	1.15	1.19	<0.0001
Hispanic	0.90	0.88	0.93	<0.0001
Asian or Pacific Islander	1.12	1.07	1.16	<0.0001
Native American	0.92	0.84	1.01	0.0793
**Characteristics of Patients**				
**Median Household Income Category for patient’s Zip code ***				
0–25th percentile	Reference			
26–50th percentile	1.03	1.01	1.04	0.0038
51–75th percentile	1.02	1.01	1.04	0.0127
76–100th percentile	1.01	0.99	1.03	0.2592
**Primary Payer**				
Medicare	Reference			
Medicaid	1.04	1.01	1.07	0.0025
Private Insurance	1.30	1.27	1.32	<0.0001
Other/Self-pay/No charge	1.46	1.42	1.51	<0.0001
**Admission type**				
Non-elective	Reference			
Elective	0.29	0.28	0.30	<0.0001
**Admission day**				
Weekday	Reference			
Weekend	1.13	1.11	1.15	<0.0001
**Characteristics of Hospitals**				
**Bed-size of hospital ^†^**				
Small	Reference			
Medium	1.11	1.09	1.13	<0.0001
Large	1.15	1.13	1.17	<0.0001
**Hospital Location & Teaching Status**				
Rural	Reference			
Urban Non-teaching	1.04	1.02	1.07	0.0019
Urban Teaching	1.14	1.12	1.17	<0.0001
**Hospital Region**				
Northeast	Reference			
Midwest	1.14	1.12	1.17	<0.0001
South	1.23	1.21	1.26	<0.0001
West	1.26	1.24	1.29	<0.0001
**Comorbidities of Patients**				
Diabetes Mellites	0.49	0.49	0.50	<0.0001
Hypertension	1.73	1.70	1.76	<0.0001
Obesity	0.83	0.81	0.84	<0.0001
Drug Abuse/Dependence	1.03	0.99	1.07	0.1261
Current Alcohol Dependence	0.91	0.88	0.94	<0.0001
Past History of Alcohol	0.83	0.70	0.98	0.0265
Current Smoker	1.30	1.27	1.32	<0.0001
Past History of Smoking	0.73	0.72	0.75	<0.0001
Acquired immune deficiency syndrome	0.12	0.10	0.13	<0.0001
Renal Failure	0.27	0.27	0.28	<0.0001
Atrial Fibrillation	1.13	1.11	1.14	<0.0001
Hemorrhagic Stroke	2.49	2.39	2.60	<0.0001
History of TIA/Stroke	1.56	1.53	1.59	<0.0001
**Deyo’s Charlson Comorbidity Index (CCI)**				
1	Reference			
2	1.86	1.83	1.89	<0.0001
3	4.39	4.30	4.48	<0.0001
4	7.39	7.22	7.57	<0.0001
≥5	9.38	9.18	9.58	<0.0001
**Area under the ROC curve/c-index**	**0.882**			

UL: Upper Limit; LL: Lower Limit; * This represents a quartile classification of the estimated median household income of residents in the patient’s ZIP code; † Bed-size of hospital indicates number of hospital beds which varies depending on hospital location (rural/urban), teaching status (teaching/non-teaching) and region (Northeast/Midwest/Southern/Western).

**Table 3 medicina-55-00475-t003:** Characteristics of patients with lipid disorders (LDs) in AIS population from January 2003–December 2014.

	LDs	Non-LDs	Total	*p* Value
**AIS (%)**	451,645 (10.69)	3,773,279 (89.31)	446,446 (100)	<0.0001
**Demographics of Patients**				
**Mean Age ± Standard Error (Years)**	70 ± 0.04	71 ± 0.01		<0.0001
**Gender (%)**				<0.0001
Female	222,900 (49.35)	1,767,703 (46.85)	1,990,602 (47.12)	
Male	228,746 (50.65)	2,005,507 (53.15)	2,234,253 (52.88)	
**Race (%)**				<0.0001
White	318,230 (72.36)	2,667,898 (72.54)	2,986,128 (72.53)	
African American	70,736 (16.08)	620,354 (16.87)	691,090 (16.78)	
Hispanic	35,048 (7.97)	276,724 (7.52)	311,772 (7.57)	
Asian or Pacific Islander	13,944 (3.17)	94,600 (2.57)	108,544 (2.64)	
Native American	1832 (0.42)	18,001 (0.49)	19,833 (0.48)	
**Characteristics of Patients**				
**Median Household Income Category for patient’s Zip code (%) ***				<0.0001
0–25th percentile	116,887 (26.40)	1,125,147 (30.48)	1,242,034 (30.04)	
26–50th percentile	105,237 (23.77)	959,992 (26)	1,065,230 (25.76)	
51–75th percentile	108,373 (24.48)	855,077 (23.16)	963,450 (23.30)	
76–100th percentile	112,216 (25.35)	751,704 (20.36)	863,920 (20.89)	
**Primary Payer (%)**				<0.0001
Medicare	290,927 (64.49)	2,532,866 (67.25)	2,823,793 (66.95)	
Medicaid	28,514 (6.32)	257,590 (6.84)	286,104 (6.78)	
Private Insurance	101,488 (22.50)	697,741 (18.53)	799,229 (18.95)	
Other/Self-pay/No charge	30,179 (6.69)	278,274 (7.39)	308,453 (7.31)	
**Admission type (%)**				0.0002
Non-elective	433,571 (96.20)	3,589,986 (95.35)	4,023,557 (95.44)	
Elective	17,130 (3.80)	175,254 (4.65)	192,384 (4.56)	
**Admission day (%)**				0.0026
Weekday	337,044 (74.63)	2,808,044 (74.42)	3,145,089 (74.44)	
Weekend	114,601 (25.37)	965,234 (25.58)	1,079,835 (25.56)	
**Characteristics of Hospitals**				
**Bed-size of hospital (%) ^†^**				<0.0001
Small	50,403 (11.19)	448,170 (11.93)	498,573 (11.85)	
Medium	115,506 (25.63)	963,139 (25.64)	1,078,644 (25.64)	
Large	284,703 (63.18)	2,345,109 (62.43)	2,629,813 (62.51)	
**Hospital Location & Teaching Status (%)**				<0.0001
Rural	41,205 (9.14)	453,884 (12.08)	495,089 (11.77)	
Urban Non-teaching	200,443 (44.48)	1,582,234 (42.12)	1,782,676 (42.37)	
Urban Teaching	208,964 (46.37)	1,720,301 (45.80)	1,929,265 (45.86)	
**Hospital Region (%)**				<0.0001
Northeast	117,433 (26)	778,923 (20.64)	896,356 (21.22)	
Midwest	74,569 (16.51)	655,617 (17.38)	730,186 (17.28)	
South	179,036 (39.64)	1,631,363 (43.23)	1,810,399 (42.85)	
West	80,608 (17.85)	707,375 (18.75)	787,983 (18.65)	
**Comorbidities of Patients (%)**				<0.0001
Diabetes	179,812 (40.02)	1,258,315 (33.50)	1,438,128 (34.20)	
Drug abuse	6735 (1.50)	82,624 (2.20)	89,359 (2.12)	
Obesity	44,246 (9.85)	286,635 (7.63)	330,881 (7.81)	
Hypertension	388,411 (86.44)	2,954,771 (78.66)	3,343,182 (79.50)	
Renal failure	45,347 (10.09)	445,605 (11.86)	490,952 (11.67)	
Acquired immune deficiency syndrome	457 (0.10)	7606 (0.20)	8063 (0.19)	
**Deyo’s Charlson Comorbidity Index (CCI)**				<0.0001
1	126,878 (28.09)	1,059,432 (28.08)	1,186,309 (28.08)	
2	110,148 (24.39)	831,212 (22.03)	941,360 (22.38)	
3	93,546 (20.71)	816,642 (21.64)	910,188 (21.54)	
4	65,014 (14.39)	537,270 (14.24)	602,284 (14.26)	
≥5	56,061 (12.41)	528,722 (14.01)	584,783 (13.84)	

* This represents a quartile classification of the estimated median household income of residents in the patient’s ZIP code; † Bed-size of hospital indicates number of hospital beds which varies depending on hospital location (rural/urban), teaching status (teaching/non-teaching) and region (Northeast/Midwest/Southern/Western). The percentage in brackets are column % indicating direct comparison between LDs vs. non-LDs amongst AIS patients.

**Table 4 medicina-55-00475-t004:** Univariate analysis of outcomes of patients with lipid disorders (LDs) among AIS admissions from January 2003–December 2014.

	LDs	No-LDs	Total	*p* Value
**Post-AIS Outcomes**				
**All Cause in Hospital Mortality (%)**	13,218 (2.93)	206,346 (5.48)	219,564 (5.21)	<0.0001
**Discharge Disposition (%)**				<0.0001
Routine/Home	187,568 (43.14)	1,299,013 (36.81)	1,486,581 (37.50)	
Transfer to Short-term Hospital	12,474 (2.87)	114,476 (3.24)	126,950 (3.20)	
Transfer to SNF/ICF/Another Type of Facility	175,573 (40.38)	1,639,408 (46.45)	1,814,981 (45.79)	
Home Health Care	59,159 (13.61)	476,296 (13.50)	535,455 (13.51)	
Discharge other than Home (%)	247,206 (56.86)	2,230,180 (63.19)	2,477,386 (62.50)	<0.0001
**APR-DRG Severity/Loss of Function (%)**				<0.0001
Minor loss of function	58,647 (13.47)	401,109 (11.32)	459,756 (11.55)	
Moderate loss of function	246,559 (56.61)	1,805,199 (50.93)	2,051,758 (51.55)	
Major loss of function	114,404 (26.27)	1,106,595 (31.22)	1,220,999 (30.68)	
Severe loss of function	15,899 (3.65)	231,623 (6.53)	247,522 (6.22)	
Major/Severe Loss of Function/Severity (%)	130,303 (29.92)	1,338,218 (37.75)	1,468,521 (36.9)	
**APR-DRG Likelihood of Death (%)**				<0.0001
Minor likelihood of death	171,426 (39.36)	1,124,229 (31.72)	1,295,655 (32.55)	
Moderate likelihood of death	195,248 (44.83)	1,644,501 (46.40)	1,839,749 (46.22)	
Major likelihood of death	54,708 (12.56)	579,523 (16.35)	634,231 (15.94)	
Severe likelihood of death	14,128 (3.24)	196,272 (5.54)	210,401 (5.29)	
Major/Extreme likelihood of death (%)	68,836 (15.8)	775,795 (21.89)	844,632 (21.23)	
**Post-AIS Complications**				
**Post-stroke early epilepsy**	21,149 (4.68)	231,487 (6.13)	252,636 (5.98)	<0.0001
**Stroke associated pneumonia**	9616 (2.13)	139,553 (3.70)	149,169 (3.53)	<0.0001
**Hemorrhagic Transformation**	5642 (1.25)	64,576 (1.71)	70,218 (1.66)	<0.0001
**Upper gastro-intestinal bleeding**	1477 (0.33)	17,152 (0.45)	18,629 (0.44)	<0.0001
**Length of Stay ± SE (Days)**	4.83 ± 0.02	5.43 ± 0.01		<0.0001
**Cost of Hospitalization ± SE ($)**	34,604 ± 154	38,547 ± 62.04		<0.0001

APR-DRG: All Patient Refined Diagnosis Related Groups; SNF: Skilled nursing facility; ICF: Intermediate care facility; SE: standard error. The percentage in brackets are column % indicating direct comparison between LDs vs. non-LDs amongst AIS patients.

**Table 5 medicina-55-00475-t005:** Multivariate logistic regression analysis of outcomes and complications in patients with LDs compared to non-LDs (reference) amongst AIS hospitalizations.

Odds Ratio	95% Confidence Interval		*p* Value	Area under the ROC Curve/c-Index
	Lower Limit	Upper Limit		
**Model 1: All cause in-hospital Mortality**				
0.66	0.62	0.69	<0.0001	0.76
**Model 2: Discharge Disposition (Home vs. no-Home)**				
0.83	0.82	0.85	<0.0001	0.76
**Model 3: APR-DRG loss of function (major/severe vs. minor/moderate)**				
0.80	0.79	0.82	<0.0001	0.82
**Model 4: APR-DRG risk of death (major/severe likelihood vs. minor/moderate likelihood)**				
0.77	0.75	0.79	<0.0001	0.81
**Model 5: Post Stroke Early Epilepsy**				
0.89	0.8	0.86	<0.0001	0.65
**Model 6: Stroke Associated Pneumonia**				
0.75	0.71	0.80	<0.0001	0.8
**Model 7: Upper GI Bleeding**				
0.85	0.73	0.99	<0.0001	0.69
**Model 8: Hemorrhagic Transformation**				
0.82	0.75	0.89	<0.0001	0.78

All models are adjusted for demographics (age, gender, race), patient-level hospitalization variables (admission day, primary payer, admission type, median household income category), hospital-level variables (hospital region, teaching versus non-teaching hospital, hospital bed-size), comorbidities, concurrent conditions like hypertension, diabetes mellitus, hypercholesterolemia, atrial fibrillation, obesity, amyloidosis, hemorrhagic transformation, smoking status, drug abuse, alcohol abuse, medication use (anticoagulant and antiplatelet medication, platelets inhibitor infusion, chronic use of aspirin), and use of IV tPA during the same hospitalization or in a different institution within the 24 h prior to admission to the facility, mechanical thrombectomy, gastrostomy, nasogastric tube insertion, invasive-noninvasive mechanical ventilation, and Charlson’s co-morbidity index (CCI).

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
