# Peer review of "The Lipid Paradox Among Acute Ischemic Stroke Patients-A Retrospective Study of Outcomes and Complications"

_medicina, 2019, doi:10.3390/medicina55080475_

Round 1

Reviewer 1 Report

Lipid disorders are one of the main vascular risk factor for cardiovascular diseases. In this retrospective study, the presence of this disorder based on ICD-9CM code is protective against post-stroke mortality, disability and complications.

However, thare are many major concerns:

It is not clear if the lipid disorders were already present at admission or diagnosed during hospitalization; this observation is critical because there are no mention concerning the use of statins that are neuroprotective and for this reason they could influence the outcome;

the mortality and disability are influenced by the acute treatment in ischemic stroke patients; for this reason it could be mandatory to add also intravenous thrombolysis and other acute stroke treatments as covariate in the multivariable analysis;

one of the main outcome is represented by disability or loss of function; it is not clear in the methodological section what is the timing of follow-up or the timing of this evaluation;

another predictive factor for the outcome of acute ischemic stroke patients is represented by clinical severity at admission, but no data were reported regarding this variable;

In the discussion paragraph, at line 261-263 - page 16, a decreased trend of lipid disorder in AIS hospitalization was justified by the etiological type of strokes; however, in this retrospective study the TOAST classification was not included and for this reason the previous hypothesis could not be considered.

Finally, please provide a language revision as well as typewriting errors (i.e. what does mean "Migraine" at the beginning of the discussion paragraph - page 16 line 236?)

Author Response

Dear Reviewer-

I appreciate your time and efforts for valuable concerns. Here is an explanation for each concern.

It is not clear if the lipid disorders were already present at admission or diagnosed during hospitalization; this observation is critical because there are no mention concerning the use of statins that are neuroprotective and for this reason they could influence the outcome;

- I have added in the revised-manuscript about LDs were identified in AIS patients as a secondary diagnosis responsible for the AIS and do not have statin utilization data, unfortunately.

the mortality and disability are influenced by the acute treatment in ischemic stroke patients; for this reason it could be mandatory to add also intravenous thrombolysis and other acute stroke treatments as covariate in the multivariable analysis;

-Table 5 modelling includes intravenous thrombolysis and other acute stroke treatments as co-variate and I will be submitting a supplemental file to show a comprehensive model.

one of the main outcome is represented by disability or loss of function; it is not clear in the methodological section what is the timing of follow-up or the timing of this evaluation; another predictive factor for the outcome of acute ischemic stroke patients is represented by clinical severity at admission, but no data were reported regarding this variable;

- Definition of the outcomes was added to the method section of the revised version. There is no timing of follow-up after discharge as outcomes decided were On-Discharge.

In the discussion paragraph, at line 261-263 - page 16, a decreased trend of lipid disorder in AIS hospitalization was justified by the etiological type of strokes; however, in this retrospective study the TOAST classification was not included and for this reason the previous hypothesis could not be considered.

- I have made changes according to your suggestion in the Discussion section. 

Finally, please provide a language revision as well as typewriting errors (i.e. what does mean "Migraine" at the beginning of the discussion paragraph - page 16 line 236?)

-Correction is made

Your feedback in draft 2 will be highly appreciated.

Thank you

Sincerely

Urvish

Reviewer 2 Report

Patel U. et al. reported that lipid paradox among acute ischemic stroke (AIS) patients from a retrospective cross-sectional study. They first showed that lipid disorders (LDs) has a higher prevalence among AIS patients and associated with patients having AIS from the NIS database at year-2014. Next, they showed that the prevalence of LDs in AIS patients decreased from 2003 to 2014. Based on the comparison of LDs and non-LDs, AIS patients with LDs have lower all-cause mortality in hospital, higher discharge disposition at home, lower APRDRG severity/Loss of function (major, severe and major/severe levels) and the likelihood of death, and post-AIS complications. Last, multivariable regression analyses showed the association of LDs with all-cause in-hospital mortality, discharge disposition, APRDRG loss of function, APRDRG risk of death, post-stroke early epilepsy, stroke associated pneumonia, upper GI bleeding, and hemorrhagic transformation. The power of this study to observe the differences between LDs and non-LDs populations is strong. However, I still have few suggestions/comments: 

1. The concluding sentence in the abstract is overstated. There is no evidence in the manuscript for “LDs in AIS is associated with improved mortality and disability”

2. Please remove “Migraine” in the first sentence of discussion (line 236) 

3. In the discussion, the authors seem to discuss the results present in the manuscript inadequately. Associations of LDs with multiple variables (table 5) may need to be discussed in the discussion section to enhance the lipid paradox among AIS patients.

Author Response

Dear Reviewer-

I appreciate your time and efforts for valuable concerns. Here is an explanation for each concern.

1. The concluding sentence in the abstract is overstated. There is no evidence in the manuscript for “LDs in AIS is associated with improved mortality and disability”

-Agreed and have not noticed that. Fixed in re-submission

2. Please remove “Migraine” in the first sentence of discussion (line 236) 

-Fixed

3. In the discussion, the authors seem to discuss the results present in the manuscript inadequately. Associations of LDs with multiple variables (table 5) may need to be discussed in the discussion section to enhance the lipid paradox among AIS patients.

- I have elaborated results in Discussion in re-submitted version.

Your feedback in draft 2 will be highly appreciated.

Thank you

Sincerely

Urvish